# HTR for Russian Empire Period Manuscripts: A Two-Stage Framework with New Annotated Resources

## Abstract

Historical handwritten documents represent a valuable source of information about the language, culture, and society of earlier periods. In the context of globalized scholarship, the development of automatic handwriting recognition tools for a wide range of languages has become increasingly important to ensure broader accessibility to the cultural heritage of different nations. Pre-revolutionary Russian presents a particular challenge for such systems due to its significant orthographic differences from the modern language. This work introduces a universal tool for recognizing handwritten documents written in pre-revolutionary Russian orthography, dated from the 19$^{th}$ century to the early 20$^{th}$ century. We present a two-stage handwritten text recognition (HTR) system combining YOLOv8-based line segmentation with TrOCR$_{pre}$, a transformer architecture pre-trained on Russian-language data. The system is performed on a manually annotated corpus of $38,501$ lines across three document types: Gubernatorial Reports ($31,083$ lines), Statutory Charters ($5,868$ lines), and Personal Diaries ($1,550$ lines), split into training, validation, and test sets. Our approach achieves a character error rate (CER) of 8.5% and a word error rate (WER) of 29.1% overall, with performance varying by document type - ranging from 4.8% CER on formal administrative documents to 19.0% CER on informal personal writings. The transformer-based architecture demonstrates a 53.8% improvement over traditional CNN-RNN baselines (from 18.4% to 8.5%), providing a practical tool for large-scale digitization of historical Russian archives. **Demo:** Interactive demo.

## 1 Introduction

Historical documents play a crucial role in preserving cultural heritage and providing the scholarly community with direct access to primary sources. These materials offer unique insights into the linguistic, cultural, and societal dynamics of earlier periods and serve as empirical foundations for studying long-term macroeconomic and political developments.

Despite their importance, the automated processing of such documents remains a substantial challenge - particularly in the case of low-resource languages. Modern optical character recognition (OCR) systems, typically powered by machine learning techniques (Garrido-Munoz et al., 2025b; Romein et al., 2025; Zhu et al., 2025), require large volumes of annotated training data. This creates a fundamental obstacle for historical languages or scripts with limited digital resources, where labeled corpora are scarce or altogether absent.

A notable case of this challenge is the recognition of handwritten Russian documents from the Russian Empire period, spanning from the 17$^{th}$ century until the orthographic reform of 1918 (Council of People's Commissars of the RSFSR, 1918b). Unlike many Western European languages - whose orthographic systems remained relatively stable - Russian orthography underwent radical changes following the 1917 revolution. The 1918 reform eliminated several letters and altered spelling rules, creating a substantial divergence between pre- and post-reform Russian texts.

By contrast, English orthography began stabilizing in the 18$^{th}$ century, especially after the publication of Samuel Johnson's dictionary in 1755 (Johnson, 1755), which contributed to enduring spelling conventions. As a result, modern OCR systems can recognize 18$^{th}$-century English texts with minimal adaptation (Edwards III, 2007; Garrido-Munoz et al., 2025a). The situation with Russian is

markedly different: the substantial orthographic transformation renders pre-reform Russian nearly a distinct variant from the perspective of automated language processing.

Our novelty lies in addressing a critical, previously unsolved gap rather than architectural innovation. Pre-revolutionary Russian (19th–early 20th century) represents a genuinely low-resource domain with no prior HTR systems or comprehensive datasets. Unlike English (stable since 18th century), Russian underwent radical 1918 orthographic reform eliminating obsolete characters (ѣ, ѳ, ѵ, і) and spelling conventions, preventing direct application of modern Russian HTR models.

According to recent estimates (Degtareva, 2022), only about 5% of archival Russian documents requiring digitization have been converted into digital form. This severely limits scholarly access to these materials and justifies classifying pre-reform Russian as a low-resource language in the context of modern text processing technologies.

Existing solutions for historical Russian OCR are typically narrow in scope. For example, the system designed for early 18th-century Petrine-era cursive targets a highly specific domain (Potanin et al., 2021), while the project focused on A.S. Pushkin's manuscripts is limited to a single individual's handwriting style (Kokorin et al., 2025). Although effective in their respective contexts, such tools are not generalizable to a broader range of documents due to the specificity of their training data and temporal focus.

To address this gap, we present *the first universal tool for handwritten text recognition (HTR) in historical Russian documents from the Russian Empire period* (19th–early 20th centuries), capable of handling diverse handwriting styles and document types within this era.

Our approach makes three key contributions:

First, we develop **a novel two-stage architecture for low-resource handwritten text recognition (Russian)**, combining YOLOv8-based layout analysis for line segmentation (Yaseen, 2024) with a specialized TrOCR transformer architecture (Li et al., 2023) for text recognition.

Second, we curate and manually annotate **the largest dataset of pre-reform Russian manuscripts to date, comprising 38,501 lines spanning diverse handwriting styles from the late 19th to early 20th centuries**. This represents the first comprehensive dataset of its kind for this historical period and writing system.

Third, **our system demonstrates state-of-the-art performance** with a character error rate (CER) of 8.5% and word error rate (WER) of 29.1% on held-out test data, substantially outperforming all existing methods in this domain.

These contributions enable, for the first time, large-scale automated digitization of historical Russian archives, opening new possibilities for digital humanities research and cultural heritage preservation.

## 2 Related Works

### 2.1 HTR for Historical Documents

Recent surveys Garrido-Munoz et al. (2025b) document the shift from CNN-RNN to transformer and LMM architectures, though complex layouts and rare scripts remain challenging. Benchmarks Romein et al. (2025); Ghaboura et al. (2025) confirm state-of-the-art performance: Humphries et al. (2025); Kim et al. (2025) achieve CER below 6 % on 18th-19th-century Western manuscripts. Self-supervised pre-training (Penarrubia et al., 2025) and post-OCR correction (Beshirov et al., 2025) further improve results, yet accuracy degrades on severely degraded or non-Latin material.

### 2.2 HTR for Pre-Revolutionary Russian Orthography

Pre-1918 Russian employed four now-obsolete letters and mandated the hard sign at word-final position, producing an alphabetic gap that prevents direct transfer of modern Russian HTR models. Limited prior work addresses only narrow subsets: Potanin et al. (2021) developed *Digital Peter* (9k lines, early-18th-century Petrine cursive) showing CRNN baselines incur triple the error on Latin datasets; Kokorin et al. (2025) focused on single-author *Chronicle of Pushkin* via hybrid OCR-plus-

rule pipeline requiring heavy manual post-editing. Neither exploits recent transformer or LMM advances.

Direct comparison with state-of-the-art models (LLMs, commercial OCR, DAN (Constum et al., 2024; Coquenet, 2025)) is methodologically invalid: modern systems lack obsolete Cyrillic characters (ѣ, ѳ, ѵ, і); DAN requires page-level annotation for Latin scripts vs. our line-level archival workflow; cross-period evaluation (Digital Peter's 18th vs. our 19th–20th century corpus) would confound results due to 150-year paleographic gap. We compare architectures trained on our corpus - the only valid approach given domain-specific constraints.

Data scarcity and orthographic divergence remain principal bottlenecks. Our system is the first generalizable approach across multiple 19th–20th century document types, enabling access to approximately 95% of undigitized Russian archival materials (Degtareva, 2022).

## 3 Corpus Description

The source material for this study comprises high-resolution color scans - digital facsimiles of handwritten documents preserved in the State Historical Archive of the Russian Federation (Russian State Historical Archive, 2024) and the Presidential Library (Presidential Library named after B.N. Yeltsin, 2024).

The corpus is characterized by a high degree of heterogeneity across multiple dimensions. From a palaeographic perspective, it features a wide range of handwriting styles and graphical conventions. Linguistically, the documents contain archaic morphological forms and diachronic orthographic variation. The physical condition of the sources varies considerably, resulting in uneven image quality. Thematically, the dataset spans a broad spectrum of genres, including private correspondence, diary entries, legal documents, and official reports. Structural layouts range from continuous prose to decorated headings and formal symbolic elements. These combined factors define a corpus of significant historical value and substantial computational complexity.

With 38,501 annotated lines across three document types, this represents the largest annotated corpus for pre-revolutionary Russian to date, featuring multi-scribe coverage spanning reports from more than 20 provinces of the Russian Empire. We are finalizing licensing with archival institutions. Immediate access is available to specialists upon request for non-commercial research via our ***Anonymous access request form***.

### 3.1 Dataset Composition

The corpus (38,501 lines) is organized into three distinct datasets, each reflecting a different type of historical document (see Fig. 1):

- **Gubernatorial Reports (19th-early 20th centuries)** – *31,083 lines*. These are official reports submitted by provincial administrations to the central government of the Russian Empire between 1804 and 1914. The content includes statistical tables, descriptive narratives, and incident records from various provinces (Razdorskii, 2011). The general records include text describing the economic and social situation of the province, as well as appendices in the form of dozens of tables with data on population dynamics, crop yields, industrial activity, education, etc. The documents exhibit a uniform clerical font, standardized terminology, and well-structured tabular formatting, but the number and structure of tables may have changed over time. Our sample includes reports from more than 20 provinces of the Russian Empire with different climatic and economic characteristics (including Arkhangelsk, Astrakhan, Moscow and Tobolsk).

- **Statutory Charters (19th century)** – *5,868 lines*. Statutory documents (charters) (19th century) – 5868 lines . These legal documents were prescribed by the Emancipation Manifesto of 1861 and record the transition of peasants from serfdom to a free state. Typically featuring hybrid layouts with handwritten clauses and printed inserts, they detail financial obligations, party lists, and administrative endorsements (Sofronenko, 1954). The stylistic formality and juridical lexicon reflect their legislative function.

- **Konstantin P. Pobedonostsev Personal Papers Collection (Diaries) (19th–20th centuries)** – *1,550 lines*. A collection of private reflections by the statesman and legal scholar

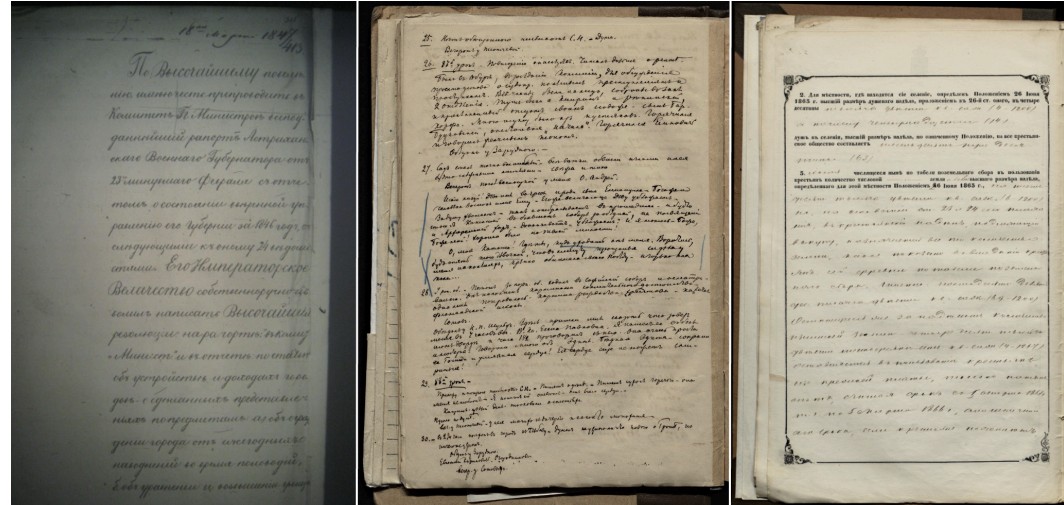

Figure 1: Example manuscript pages from the three datasets used in this study: Gubernatorial Reports (left), Personal Diaries of Konstantin P. Pobedonostsev (center), and Statutory Charters (right). The examples illustrate the diversity of handwriting styles, document structure, and historical orthography encountered across different sources.

who oversaw educational and ecclesiastical reforms. The texts offer insights into everyday life, courtly affairs, and personal impressions. Written in a freeform style, they exhibit emotional interjections, nonstandard constructions, orthographic irregularities, and extensive manual corrections.

While the first two datasets reflect official documentation in polished chancery handwriting, the third represents informal, idiosyncratic entries marked by irregular lineation, overwrites, and expressive variation. The complete dataset is currently being prepared for public release to support future research in historical document digitization.

### 3.2 Linguistic Specificity and Representativeness

All three datasets belong to a highly specific linguistic class characterized by pre-reform Russian orthography, archaisms, and palaeographic idiosyncrasies. These factors impose additional demands on HTR systems and necessitate domain expertise during annotation (see Appendix A).

The corpus includes both formal administrative genres and spontaneous private writing, thus capturing a broad spectrum of textual production practices in Imperial Russia. Despite being geographically limited to archival collections, the documents span a representative array of Eastern European written forms from the 19th to early 20th centuries. Stylistic and graphic features include syntactic complexity, orthographic variability, nonstandard abbreviations, and a wide range of handwriting - from chancery calligraphy to diary informality.

In total, over 38,000 lines were transcribed by domain experts, including historians trained in Old Church Slavonic and pre-revolutionary Russian script. This expert-driven annotation is critical given the rare and often ambiguous glyphs present in the sources.

### 3.3 Annotation Format and Procedure

The dataset partitioning follows standard machine learning practices with training, validation, and test splits designed to ensure robust model evaluation:

*Gubernatorial Reports (19th–early 20th centuries) – 31,083 lines:*

- Train: 28,671 lines (92.2%)
- Validation: 1,191 lines (3.8%)
- Test: 1,221 lines (3.9%)

*Statutory Charters (19th century) – 5,868 lines:*

- Train: 4,441 lines (75.7%)
- Validation: 662 lines (11.3%)
- Test: 765 lines (13.0%)

*Konstantin P. Pobedonostsev Personal Papers Collection (Diaries) (19th–20th centuries) – 1,550 lines:*

- Train: 960 lines (61.9%)
- Validation: 264 lines (17.0%)
- Test: 326 lines (21.0%)

The training sets were used for model parameter optimization, validation sets for hyperparameter tuning and early stopping criteria, and test sets for final performance evaluation. The varying split ratios reflect the different dataset sizes, with smaller collections allocated proportionally larger validation and test portions to ensure statistically meaningful evaluation metrics.

The annotation process was conducted using the LabelStudio (Heartex, 2020) platform and followed a two-stage pipeline: (1) line localization and (2) content transcription.

**Stage 1: Line Localization.** Two complementary techniques were employed depending on the structural complexity of the source material:

- **Bounding-box detection** was applied to documents with regular layouts and well-separated lines, specifically the *Gubernatorial Reports* and *Statutory Charters*. Each text line was enclosed in an axis-aligned rectangle.
- **Segmentation masks** were used for the *Personal Diaries*, which often featured overlapping, skewed, or irregularly spaced lines. This approach enabled finer-grained delineation of visually entangled handwriting.

**Stage 2: Line Transcription.** Each localized line - regardless of whether it was detected via bounding box or segmentation - was manually transcribed by expert annotators. The final annotated unit thus consists of a paired representation: a cropped image of the handwritten line and its corresponding textual content in UTF-8 format.

## 4 Architecture of the Recognition System

Our system adopts a two-stage architecture comprising YOLOv8-based line segmentation (Yaseen, 2024) and TrOCR transformer-based text recognition (Li et al., 2023), trained on three manually annotated datasets of late 19[th] to early 20[th] century manuscripts (totaling 38,501 lines). The complete pipeline implements a four-stage process: image preprocessing, text line localization, optical character recognition, and postprocessing This modular design ensures high accuracy under challenging paleographic conditions.

### 4.1 Image Preprocessing

Each image undergoes a series of enhancement operations to optimize the quality of the input prior to recognition. The preprocessing stage includes:

- noise suppression to remove background artifacts;
- contrast enhancement to improve text-to-background visibility.

Empirical evaluations confirm consistent accuracy gains across all tested architectures. While modern HTR models may be less sensitive to preprocessing on high-quality contemporary documents, our degraded historical manuscripts-exhibiting ink deterioration, paper discoloration, and variable contrast-demonstrably benefit from these enhancement operations. Preprocessing is therefore applied both during model training and inference.

## 4.2 Text Line Localization

A modified YOLOv8 (Yaseen, 2024) performs joint detection-and-segmentation for localizing handwritten lines, enabling precise identification of overlapping or distorted text. Trained on challenging configurations (overlapping, slanted, curved lines), YOLOv8 employs mask-based cropping for complex layouts: an axis-aligned bounding box is extracted with pixels outside the segmentation mask set to neutral background, preserving original geometry without explicit warping. Selected after benchmarking against alternative solutions (Tesseract OCR, EasyOCR, Craft Text Detector) that proved unreliable on degraded manuscripts, YOLOv8 achieves high localization accuracy (mAP $= 0.98$) on the validation set.

## 4.3 Text Recognition

Following localization, cropped line images are passed to the OCR component. We evaluate five architectures: *three traditional baselines* (**VGG-CTC** (Simonyan & Zisserman, 2015; Graves et al., 2006), **CRNN**(Shi et al., 2015), **ResNet-BiLSTM-CTC**(Shonenkov et al., 2021)) and *two transformer-based models* (**TrOCR**, **TrOCR$_{pre}$**) (Li et al., 2021). Among these, TrOCR$_{pre}$ consistently delivered the highest accuracy and was selected for the final pipeline due to its robust handling of non-horizontal text through grid-like feature maps (detailed evaluation in Section 6.1).

Among these, TrOCR$_{pre}$ consistently delivered the highest character-level accuracy and was selected for downstream integration. The transformer architecture's grid-like feature maps enable robust decoding of non-horizontal text. Through exposure to diverse line geometries during fine-tuning-including slanted, curved, and irregular alignments prevalent in Personal Diaries: TrOCRpre learns to decode text along varied spatial trajectories without requiring explicit line straightening.

## 4.4 Postprocessing

To improve transcription quality, two postprocessing strategies were evaluated:

1. **Dictionary-based correction**:

   - a lexicon was constructed from the training corpus;
   - Levenshtein-distance matching was used to suggest corrections;
   - improvements were observed only on subsets with initially low accuracy, showing limited overall benefit.

2. **LLM-assisted correction**:

   - the model received the erroneous OCR output and reference context;
   - an LLM generated a corrected version based on semantic coherence;
   - performance consistently degraded due to inappropriate historical substitutions: modern LLMs lack training data in pre-reform orthography; their tokenizers and corpora do not contain obsolete characters (ѣ, ѳ, ѵ, і) or archaic spelling conventions, leading to undesirable modernization of historical text. Integrated language modeling with historical Russian corpora represents a promising direction for future work.

# 5 Training and Inference Procedure

## 5.1 Training Pipeline

The training process followed a sequential pipeline composed of the following stages:

1. **Image preprocessing:** All images were enhanced using the standard preprocessing routine described in Section 3 prior to any downstream training.

2. **Data partitioning:** The dataset was split into three subsets - train, validation, and test - at the document level. Pages originating from the same source document were never assigned to multiple subsets, thereby preventing content leakage.

3. **Detector and segmenter training:** The YOLOv8 architecture was trained on annotated data using both bounding-box and segmentation-mask formats. This dual-format annotation enabled the model to generalize across both well-structured and visually entangled layouts.

4. **Text recognition training:**

   - Line images were extracted from the localized regions.
   - Each cropped line was paired with its corresponding transcription and used to train OCR models, as detailed in Section 3.3.

5. **Postprocessing optimization:** For each postprocessing strategy, hyperparameters were tuned to maximize evaluation metrics on the validation set. This included thresholding for dictionary-based correction and alignment strategies for candidate selection.

## 5.2 Inference Workflow

During inference, the system operates as a unified pipeline:

1. Each input page is passed through the image preprocessing module.

2. YOLOv8 performs line localization using either bounding boxes or segmentation masks, depending on the document type.

3. Localized line images are processed by the chosen OCR model (e.g., TrOCR$_{pre}$).

4. Optional postprocessing is applied to refine the textual output.

## 5.3 Model Adaptation and Fine-Tuning

To tailor the recognition pipeline to the characteristics of pre-revolutionary Russian manuscripts, both detection and recognition components were adapted:

- **YOLOv8** was fine-tuned on historical documents annotated to capture the specificities of early Cyrillic script, including characters with diacritical marks and obsolete graphemes.

- **TrOCR$_{pre}$** was initialized from pretrained weights and further fine-tuned on the task-specific corpus. The adaptation included additional exposure to synthetic Russian strings.

# 6 Experiments and Evaluation

## 6.1 Comparison of HTR Architectures

To evaluate handwriting recognition performance, we conducted experiments with five model architectures:

1. **VGG-CTC**: a CNN architecture with CTC decoding (based on PyLaia);

2. **CRNN**: a hybrid model combining CNN and RNN layers with a CTC decoder (adapted from wronnyhuang/htr);

3. **ResNet-BiLSTM-CTC**: deep convolutional layers followed by bidirectional LSTMs (from the indic-htr project);

4. **TrOCR**: a transformer-based model with a Vision Transformer encoder, pre-trained on English handwriting corpora;

5. **TrOCR$_{pre}$**: the same architecture as TrOCR, further pre-trained on synthetic Russian lines and fine-tuned on real handwritten Russian and Kazakh data.

We do not include general-purpose OCR engines as recognition baselines. Their default Russian models target modern orthography and printed text and lack coverage of obsolete graphemes; without retraining or custom alphabets they produce inconsistent or modernized outputs on pre-reform handwriting, making results non-comparable. We therefore report only HTR architectures trained on our corpus; off-the-shelf OCR was used, if at all, only for line-detection sanity checks.

The experimental results, summarized in Table 1, demonstrate a clear advantage of transformer-based models. The best overall performance was achieved by the **YOLOv8 + TrOCR$_{pre}$** configuration, reaching a CER of 8.5% across the full corpus. More granular metrics per document type are presented in Table 2, where the same configuration achieved a CER of 4.8% on the *Gubernatorial Reports* subset. Examples of system usage can be found in Appendix B.

### 6.2 Component-wise Evaluation

#### 6.2.1 Image Preprocessing

In ablations, image preprocessing improved both line localization and transcription accuracy. Because filtering operations (noise suppression, contrast enhancement) yielded consistent gains on the validation set, we include them in the final pipeline.

#### 6.2.2 Line Detection and Segmentation

We compared our detection module against standard tools (Tesseract OCR, EasyOCR, CraftTextDetector). These baseline tools proved unstable when applied to historical manuscripts. In contrast, the fine-tuned YOLOv8 model unified detection and segmentation stages and achieved high localization accuracy (mean average precision = 0.98). Additional evaluation on the test set confirms YOLOv8's robustness, with mean IoU of 97.2%, indicating precise line boundary delineation even in challenging layouts.

Cross-archive out-of-distribution evaluation is currently impossible: this represents the first publicly annotated dataset for pre-revolutionary Russian from this historical period. However, we demonstrate multi-scribe generalization capability on the Gubernatorial Reports subset, which contains documents from more than 20 provinces of the Russian Empire, each written by different scribes with varying handwriting styles. We are actively collaborating with archival institutions to create annotated manuscripts from other archives, which will enable rigorous cross-archive benchmarks in future work.

To isolate the contribution of each pipeline component, we conducted experiments using manually annotated ground-truth bounding boxes and segmentation masks. Under these ideal localization conditions, TrOCRpre achieved 7.2% CER compared to 8.5% CER in the full end-to-end pipeline, yielding a segmentation-to-recognition error gap of only 1.3 percentage points. This indicates that while line detection errors do propagate to the final output, the recognition architecture remains the dominant factor in overall performance.

Furthermore, TrOCRpre demonstrates robustness to non-horizontal text geometries without requiring explicit line straightening or geometric warping. Through exposure to diverse line orientations during fine-tuning-including slanted, curved, and irregular alignments prevalent in the Personal Diaries subset-the model learns to decode text along varied spatial trajectories. The achievement of 19.0% CER on the most challenging Personal Diaries corpus, despite its extreme geometric irregularity, empirically validates this geometric robustness.

#### 6.2.3 Postprocessing

Postprocessing experiments yielded the following observations:

- Dictionary-based correction helped in cases with low initial recognition quality but slightly degraded performance when the base output was already accurate.
- Integration of large language models (LLMs) in the current setup led to distortion of the original historical content and requires further investigation.

### 6.3 Final Results

The final CER and WER metrics reflect the cumulative error of the two-stage pipeline: line localization and text recognition. As shown in Table 1, the **YOLOv8 + TrOCR$_{pre}$** system achieved:

- CER = 8.5% and WER = 29.1% across the full corpus test-subset (2,312 lines), as detailed in Table 1,

- CER = 4.8% and WER = 21.5% on the Gubernatorial Reports test-subset (1,221 lines), as detailed in Table 2.

To evaluate the recognition stage independently from detection quality, we conducted additional experiments using manually annotated bounding boxes and segmentation masks. Under these ideal conditions, **TrOCR$_{pre}$** achieved a CER of 7.2% and WER of 24.0%, highlighting the architecture's strong potential when accurate localization is provided.

Table 1: Overall CER and WER on the full corpus test-subset (2,312 samples)

| Model | CER (%) | WER (%) |
|---|---|---|
| VGG-CTC | 18.4 | 50.3 |
| CRNN | 12.8 | 39.8 |
| ResNet-BiLSTM-CTC | 9.8 | 32.2 |
| YOLOv8 + TrOCR | 9.3 | 30.2 |
| YOLOv8 + TrOCR$_{pre}$ | **8.5** | **29.1** |

Table 2: CER and WER across individual test-subsets: *Gubernatorial Reports*, *Statutory Charters*, and *Personal Diaries*

| Model | Gubernatorial Reports | | Statutory Charters | | Personal Diaries | |
|---|---|---|---|---|---|---|
| | CER | WER | CER | WER | CER | WER |
| VGG-CTC | 10.4 | 36.2 | 20.2 | 54.4 | 28.8 | 72.4 |
| CRNN | 7.9 | 30.1 | 13.1 | 42.3 | 24.5 | 63.2 |
| ResNet-BiLSTM-CTC | 5.5 | 24.1 | 11.2 | 33.7 | 20.5 | 54.2 |
| YOLOv8 + TrOCR | 5.2 | 23.5 | 10.5 | 32.6 | 20.1 | 53.5 |
| YOLOv8 + TrOCR$_{pre}$ | **4.8** | **21.5** | **9.4** | **30.8** | **19.0** | **50.0** |

## 7 Results and Discussion

Our evaluation on 2,312 test samples shows that transformer-based models consistently outperform traditional CNN–RNN architectures for historical Russian HTR. The YOLOv8 + TrOCR$_{pre}$, configuration achieved the lowest error rates (CER: 8.5%, WER: 29.1%), representing a 53.8% reduction in CER compared to the VGG-CTC baseline (18.4% CER) as shown in Table 1.

While a WER of 29.1% overall (50% on diaries) indicates that manual correction remains necessary, this represents meaningful progress for an extremely challenging domain. Importantly, error analysis reveals that many mistakes affect functional words or minor orthographic variants while preserving key entities and overall semantic structure. This characteristic significantly reduces the amount of meaningful human correction needed compared to the baseline errors produced by traditional architectures, making the output practically useful for search, reading, and further scholarly analysis.

The performance hierarchy follows a clear pattern: VGG-CTC < CRNN < ResNet-BiLSTM-CTC < TrOCR < TrOCR$_{pre}$, with each architectural advancement yielding measurable improvements. Notably, language-specific pretraining (TrOCR$_{pre}$ vs. TrOCR) provides consistent gains across all document types, with improvements of 0.4-1.1% CER as evidenced in Table 2.

### 7.1 Document Type Impact on Recognition Accuracy

Recognition performance varies systematically across document types, correlating with their structural and linguistic characteristics (Table 2). Gubernatorial Reports achieve the highest accuracy (4.8% CER, 21.5% WER), followed by Statutory Charters (9.4% CER, 30.8% WER), and Personal Diaries (19.0% CER, 50.0% WER) using the best-performing configuration.

Performance correlates strongly with writing formality: 4.8% CER on formal administrative reports versus 19.0% CER on informal personal diaries. This fourfold accuracy gap reflects not only the inherent task difficulty posed by irregular handwriting styles but is also compounded by severe data scarcity: Personal Diaries comprise only 960 training samples compared to 28,671 samples for Gubernatorial Reports. The CER-to-WER ratio also differs by type: 1:4.5 for formal documents vs. 1:2.6 for personal diaries-consistent with shorter word lengths and greater lexical diversity in informal writing.

## 7.2 Training Data Requirements

Dataset size directly impacts model performance, particularly for complex document types. Personal Diaries, with only 960 training samples, show the highest error rates (19.0% CER) despite representing the most challenging recognition task. In contrast, Gubernatorial Reports with 28,671 training samples achieve optimal performance (4.8% CER). This performance gap reflects the inherent scarcity of preserved informal historical manuscripts from the Russian Empire period rather than methodological limitations in dataset construction, as data scarcity compounds the inherent difficulty of irregular handwriting styles.

## 7.3 Component Contribution Analysis

Ablation studies reveal that accurate line localization is critical for overall system performance. When evaluated with manually annotated bounding boxes, TrOCR$_{pre}$ achieves 7.2% CER compared to 8.5% CER in the full pipeline, indicating that detection errors contribute 1.3 percentage points to the total error rate. On the validation set, the YOLOv8 model attains mAP = 0.98, indicating high detection performance on this corpus.

Image preprocessing provides consistent improvements across all architectures, while postprocessing methods show mixed results. Dictionary-based correction helps only when base accuracy is low, and LLM-based correction degrades performance due to inappropriate historical text modifications.

This analysis establishes quantitative baselines for historical Russian HTR and shows that transformer architectures with language-specific pretraining achieve the lowest CER/WER among the evaluated models.

# 8 Conclusion

This study demonstrates that modern computer vision and natural language processing techniques can be successfully adapted for handwritten text recognition (HTR) in historical documents of varying types. We acknowledge using established components (YOLOv8 + TrOCRpre). However, the contribution lies in the successful adaptation and comprehensive evaluation of these architectures for this unique domain-pre-revolutionary Russian manuscripts from the 19th to early 20th centuries. Our system demonstrates a 53.8% CER improvement over traditional CNN-RNN baselines (from 18.4% to 8.5%), establishing the first viable automated processing pipeline for this previously inaccessible historical corpus.

The best results were obtained using transformer-based architectures, particularly when combined with language-specific pretraining and accurate line localization.

Recognition quality varied with the typology of the historical material. Official documents written in calligraphic script achieved high accuracy (CER as low as 4.8%, WER 21.5%), whereas Personal Diaries with individual handwriting styles were more challenging (CER up to 19%, WER 50.0%). These findings highlight the need for a differentiated processing approach tailored to document type, as well as the importance of constructing representative and typologically diverse training corpora.

The resulting system achieves a character error rate (CER) of 8.5% and a word error rate (WER) of 29.1% on a held-out test set, consistently outperforming other approaches across all three document types (Gubernatorial Reports, Statutory Charters, and Personal Papers) in this domain and enabling large-scale digitization of historical Russian archives.

The results establish a solid foundation for advancing automated processing of historical archival materials. Furthermore, they open new opportunities for digital humanities research by enabling scalable access to large volumes of previously unstructured handwritten sources.

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

## A    Orthographic Differences in Pre- and Post-Reform Russian

The 1918 Russian orthography reform was a pivotal event that fundamentally reshaped written Russian, creating distinct visual and linguistic characteristics between pre-reform (prior to 1918) and post-reform (after 1918) texts (Council of People's Commissars of the RSFSR, 1918a; Kuznetsov; Kharitonova; Shapovalova). These changes are crucial for understanding the challenges in historical Handwritten Text Recognition (HTR). Key Differences:

1. **Elimination of Obsolete Letters**: Four letters, primarily serving etymological rather than phonetic purposes, were removed from the alphabet:
   - ѣ (Yat): Replaced by Е. Its usage often required memorization as its pronunciation had largely merged with 'Е'.
   - і (Decimal I): Replaced by И. Used before vowels, the hard sign, and the letter Й (e.g., исторія).
   - ѳ (Fita): Replaced by Ф. Primarily used in words of Greek origin (e.g., ѳеатръ).
   - ѵ (Izhitsa): Replaced by И or В. Used very rarely, mainly in certain Greek borrowings (e.g., мѵро).

2. **Abolition of the Hard Sign (Ъ) at the End of Words**: Prior to the reform, the hard sign Ъ was obligatorily written at the end of every word ending in a hard consonant, with no phonetic value (e.g., домъ, городъ). The reform eliminated this practice, retaining Ъ only as a separator between a prefix and a root (e.g., объяснить, подъём). This significantly altered the visual density and appearance of texts.

3. **Simplification of Prefix Spelling**: Rules for prefixes ending in -З/-С before voiced/voiceless consonants were largely streamlined. For instance, prefixes like без-, воз-, из-, раз- consistently ended in З before voiced consonants and vowels, and in С before voiceless consonants. While these rules existed before, the reform clarified and solidified their application.

4. **Other Minor Changes**: The reform also included adjustments to the spelling of compound nouns, certain grammatical endings, and a general move towards phonetic principles over etymological ones.

These fundamental changes resulted in distinct visual forms and vocabulary, making pre-reform Russian texts a highly specialized domain for HTR. Models trained on modern Russian or other languages without adaptation for these obsolete characters and spelling rules are largely ineffective, underscoring the necessity and novelty of our dedicated approach.

## B    Recognition Results

Here we can look at examples of documents and their transcripts in text form.

Listing 1: Results of text recognition on Figure 2

Николаевской
Железней дороги, изъ дровянскаго насажденяі и кустарныхъ зарослей удельной лесной дачи, такое же точно количество величи девятнадцать бесятинъ, тысяча шириста сажень (19-1300), безъ высоканяі планы ки лесной материале на ономъ. О том, те, годъ таковое реверстанеі произвести, оболее ного на выкотировке съ планшета, которая бомъ быть передана 8.7 Мировому Посре нику. Но независимо сего, на основаніи 49 ст. упомянутаго Положения, крестьяне открыя щеі отъ нихъ участки степаняются в своего пользовании по 5 Марта 1866 года, безъ всякой пото платы, а въ те время должно быть исполнено въ натуре означенное здесь развер ственеі, по назначеннымъ до согласяі кре стьянъ наризкамъ. После сего ссаже предел не пожелокъ, то смотре изъ участки сохранить въ своемъ пользовании, на не ино не, какъ по особому добровольном того дору съ удельнымъ Солдатствомъ, вовсякоми жащихъ оныя должны отойдели изъ Крестьянскаго визделяі и поступить

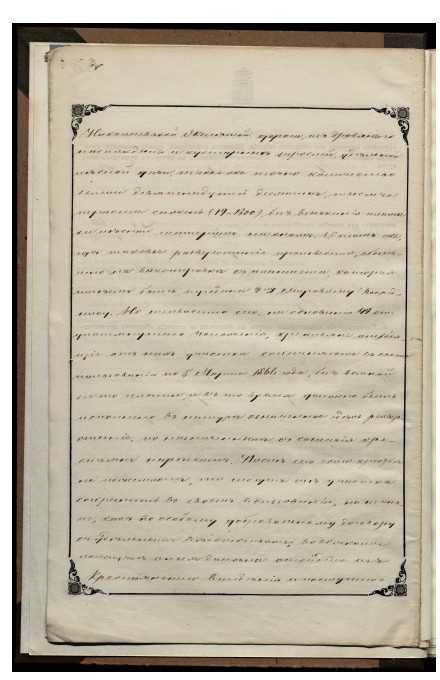

Figure 2: Document example.

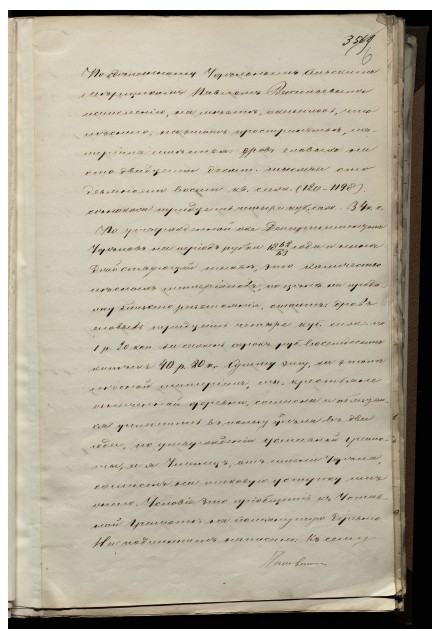

Figure 3: Document example.

Listing 2: Results of text recognition on Figure 3

По
делебному Удельнымъ Семьским мерщикомъ Павлом Васильевымъ начислению, на месте, оказалось, что леснаго, надимыхъ пространствъ, ма териала имеется: дровъ словыхъ на сто двадцати десят. тысяча ста девяносто восми кв. саж. (120-1198) сенокоса тридцать четыре куб. саж..34 к. ю. по утвержденной же Департаментомъ Уделовъ на периодъ рубки 1863 года и ныне действующий таксе, что количестьо лесныхъ материаловъ, по цене на прода ону близкаго разстояняі, места. дровъ словыхъ тридцать четыре куб. сесок. то 1 р. 20 коп. на сажень сорокъ руб. восемьдесять количенъ 40 р. 80 к. Одину, на этомъ лесной матераlьі, мы, крестьяне означенной деревни, согласны и обязуем ся уплатить въ пользу удела в два года, по утверждении уставной грамо ты, а я Уманецъ, отъ имени Удела, согласенъ на таковую уступку имъ онаго. Условие это приобщить къ устав ной грамоте на помянутую деревню На подлинномъ написано: Къ сему

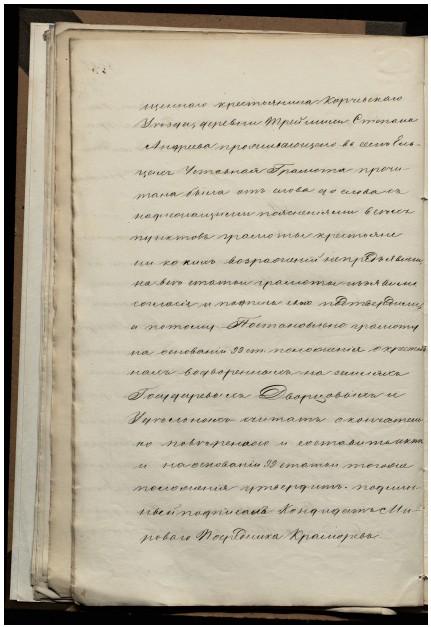

Figure 4: Document example.

Listing 3: Results of text recognition on Figure 4

щенного
крестьянина Корчевскаго Уезда, деревни Трейшихи Степана Андреева проживающего въ семъ Ель цахъ Уставная Грамота прочи тана была отъ слова до соловыхъ надлежащими поясненямиі, всехъ пунктовъ грамоты крестьяне ни какихъ возраженйі непредъявили, на все статьи грамоты изъявили согласеі и подпись ева по детвердили, а потому Постановлено грамоту на основании 99 ст. положеняі о крестья нахъ водворенных на землях Государевъ хъ Дворцовича и Удельныхъ считать окончателъ но поверенною и составителяхъ. и на основаниі 99 статьи того же положения утвердить подлин ный подписалъ: Кондидатъ Ми роваго Посредника Краморевъ.

