# OpenReview forum: "HTR for Russian Empire Period Manuscripts: A Two-Stage Framework with New Annotated Resources"
_ICLR.cc/2026/Conference — Submitted to ICLR 2026_

### Official Review · Reviewer_ekgW · 2025-10-30

**Soundness:** 2
**Presentation:** 2
**Contribution:** 1
**Rating:** 2
**Confidence:** 4

**Summary:**

The article proposes a two-stage HTR pipeline: YOLOv8 for line detection/segmentation + TrOCR for recognition, with a Russian-centric pretraining variant (TrOCRpre). Reports mAP = 0.98 for line localization and CER = 8.5%, WER = 29.1% overall (best CER 4.8% on formal documents; 19.0% on diaries).  It curates a new annotated corpus of 38,501 lines from Russian Empire–period sources (Gubernatorial Reports, Statutory Charters, Personal Diaries), with train/val/test splits per subset.

**Strengths:**

1. A sizeable, expert-annotated corpus (38,501 lines) spanning three distinct genres and writing styles—this alone is a meaningful contribution for a low-resource historical script with pre-1918 orthography (obsolete graphemes, final hard sign, etc.). The paper details composition and splits and explains the annotation workflow (bbox vs mask for irregular diaries).

    2. They tried dictionary correction (sometimes helps) and LLM-assisted correction (makes historical text worse). Reporting the negative LLM result is valuable signal for the community.

**Weaknesses:**

1. YOLO (even YOLOv8) is built for object detection, i.e., instances with:
    • Discrete boundaries,
    • Non-overlapping regions,
    • Relatively uniform aspect ratios,
    • Box-like shapes.
Handwritten text lines, however, violate all of those assumptions: But from the text it is not clear why Yolo could work in a situation where two consecutive text lines are heavily overlapped !!!

    2. Comparability to external baselines is limited: The paper does not compare the proposed method with Existing baseline method like DAN published in PAMI2023 and several other SOTA methods published in the recent past.  Even those SOTA methods are meant for some other scripts, their codes are mostly available in GITHUB, so the authors could have easily conducted experiments on their data for a fair comparison.

    3. WER remains high for downstream usability: Even with the best setup, WER = 29.1% overall and 50% on diaries implies heavy manual correction.
    4. YOLOv8 reference ambiguity:  The text credits YOLOv8 but cites Redmon et al., 2016 (YOLOv1). This should be clarified/corrected for scholarly accuracy.

**Questions:**

a) Could you add line-level recall/precision, merge/split/trim error breakdown, and show how these errors propagate to CER/WER with a few qualitative examples?

b) Could you evaluate cross-archive performance (unseen collections), cross-writer splits for diaries, and data-efficiency curves to show sample complexity by genre.

---

### Official Review · Reviewer_aiU9 · 2025-10-31

**Soundness:** 2
**Presentation:** 3
**Contribution:** 2
**Rating:** 4
**Confidence:** 4

**Summary:**

This paper details a two-stage process for handwritten text recognition (HTR) of Russian Empire-period handwritten manuscripts. The authors perform experiments with 38,501 lines extracted from three types of Russian language documents, including legal/administrative documents and personal diary pages.  They fine-tuned different models based on the traditional HTR systems (CNN-RNN-CTC combo or variants)  and the modern transformer-based architectures (TrOCR). The authors reported an overall character error rate (CER) of 8.5% with a word error rate (WER) of 29.1%. They reported that a pretrained transformer on synthetic Russian data and Kazakh handwritten lines had the best performance.

**Strengths:**

1. A full-page HTR system for historic Russian documents has been developed with a low CER on formal administrative/legal Russian documents.
2. The annotated and transcribed dataset created by the authors would be very helpful to the research community working on HTR and NLP for pre-reform (1918) Russian language. I would encourage the authors to release their dataset.
3. HTR has been evaluated on different types of models.

**Weaknesses:**

1. The paper lacks novelty as the authors trained/fine-tuned already existing models on their dataset.
2. It would be nice to include a section in the appendices that gives a brief summary of the differences in letters from the pre- and post-reform Russian texts.
3. The authors have a separate step for preprocessing images. Were any image augmentation techniques tried during training to eliminate the need for the preprocessing stage?
4. It would be nice to include a proper ablation results table in the paper that clearly shows the component-wise contribution of each phase. I do not see this in the main text or the appendices.
5. The authors mention that either bounding boxes (axis-aligned rectangles) or segmentation masks were used to detect textlines. There is no mention of warping the textlines to a horizontal rectangular space to cater for slanted or vertical text lines (normally margins are written vertically). Would this not severely restrict HTR performance where the lines are slanted, vertical, or upside down?
6. It would be nice to include the CER/WER of existing HTR systems on Russian datasets mentioned in Section 2.2.
7. In the appendices, it would be helpful to highlight the substitution, insertion, and deletion errors in the predicted text. Otherwise, these results cannot be meaningfully understood by non-Russian speakers.

Minor

8. Please start the sentence with capital letters in the bulleted lists on page 6.
9. Page 6, image preprocessing is not described in Section 3 (should be Section 4).

**Questions:**

1. What is the final output of the model? Is this word-based or character-based HTR?
2. How much improvement do you get by including the preprocessing and postprocessing steps?
3. When would you be able to release the dataset?
4. Would the HTR work well on very slanted textlines when segmentation masks are used? (This is a repeated point from weaknesses.) The authors mention that either bounding boxes (axis-aligned rectangles) or segmentation masks were used to detect textlines. There is no mention of warping the textlines to a horizontal rectangular space to cater for slanted or vertical text lines (normally margins are written vertically). Would this not severely restrict handwritten pages where the lines are slanted?

**Details Of Ethics Concerns:**

If the dataset is released, do the authors have permission to release the images?

---

### Official Review · Reviewer_BcVz · 2025-11-01

**Soundness:** 2
**Presentation:** 2
**Contribution:** 2
**Rating:** 2
**Confidence:** 4

**Summary:**

This work presents a two-stage handwritten text recognition (HTR) system for low resource language documents, in particular pre-revolutionary Russian documents. The proposed method combines YOLOv8-based line segmentation with a transformer model (TrOCRpre) pre-trained on Russian-language data. A side contribution is a corpus of over 38,000 annotated lines across three document types. Trained on this corpus, the system achieves a good overall performance (CER: 8.5%, WER: 29.1%), with notable gains on formal texts. The transformer-based approach shows a 53.8% improvement over traditional CNN-RNN baselines, offering a practical solution for large-scale digitization of historical Russian archives.

**Strengths:**

This work addresses a historically and linguistically significant challenge by focusing on handwritten Russian documents from the 19th and early 20th centuries—a seminal contribution to low-resource and historical language processing.

It introduces a novel, manually annotated dataset of over 38,000 lines across diverse document types, which will be valuable not only for the Russian HTR community but also for broader multilingual and historical handwritten text recognition research.

The system demonstrates good performance and a substantial improvement over traditional baselines, showcasing the effectiveness of transformer-based architectures in historical document digitization.

**Weaknesses:**

The proposed architecture is constructed based on standard off-the-shelf components (YOLOv8 + TrOCRpre configuration). The methodological contribution is limited, and the work is rather a study on the performance of different baselines, to demonstrate the effectiveness of Transformer architecture rather than a novel method.

The study of state of the art is limited. Understanding that it is not extensive on historical Russian handwritten documents, a more exhaustive review on historical HTR systems and subsequent evaluation would improve the quality of the paper.

The writing style could benefit from refinement. At times, the presentation resembles a technical report or proof-of-concept documentation more than a polished scientific paper, which may affect readability and clarity for a broader research audience.

The experimental evaluation, while promising, appears somewhat limited in scope. Expanding the range of experiments or including additional comparative baselines specifically designed for HTR of historical documents (for example, techniques designed for other languages, and considering domain transfer, or fine tuning strategies) could strengthen the empirical support for the proposed approach.

**Questions:**

Please see the comments provided in previous sections.

The dataset includes reports from over 20 provinces of the Russian Empire, which suggests a potentially rich diversity in handwriting styles. To better assess the recognition challenge, it would be helpful to know the number of distinct writers or writing styles represented in the corpus. I couldn’t find this information in the paper.

A minor error, the following reference is repeated: Carlos Garrido-Munoz, Antonio Rios-Vila, and Jorge Calvo-Zaragoza. Handwritten text recognition: A survey, 2025b. URL https://arxiv.org/abs/2502.08417.

---

### Official Review · Reviewer_PNbe · 2025-11-03

**Soundness:** 1
**Presentation:** 1
**Contribution:** 1
**Rating:** 0
**Confidence:** 5

**Summary:**

This article presents a system for transcribing handwritten documents in Russian, consisting of pre-processing, a line detector (YOLO), a handwriting recognition model (TrOCR) adapted to Russian, and post-processing. The system is trained and evaluated on a dataset consisting of three sources (38,000 lines).

**Strengths:**

The article focuses on document recognition in Russian, a language with few resources for model training and for which the performance of current OCR/HTR models is very poor.

**Weaknesses:**

- The article presents a system that is merely a classic combination of a Yolo-based line detector and a TrOCR model adapted to Russian.  The level of novelty is very low.
- The data is not public and therefore does not allow for reproducibility of the results.
- The proposed model is not compared to any other state-of-the-art model, and in particular to any transformer-based model or LLM.
- The model is not tested on any public dataset (e.g. Digital Peter), which would allow it to be compared to existing models.
- The section on related work consists only of surveys, and no references are given to the most efficient or standard models currently available, apart from TrOCR.

**Questions:**

L067 : "to address this gap, we present the first universal tool for handwritten text recognition (HTR) in historical Russian documents from the Russian Empire period " : There is nothing universal about the proposed model: it is trained and evaluated on a single, limited corpus without any out-of-distribution evaluation.

L177 : The complete dataset is currently being prepared for public release to support future research in historical document digitization : If the dataset is not publicly available, there is no guarantee that it ever will be.

L254 Empirical evaluations confirmed that preprocessing...  : Pre-processing usually has no significant impact on modern HTR models. These empirical evaluations must be presented.

Section 6.1: No references are given for the five models tested and no details are provided on how they are trained.

L379 : Post processing does not help : It is well known that this kind of post-processing, when used as a post-correction, is ineffective: it must be incorporated as a language model, combining the probabilities of the recogniser and the language model, as well as an optimised combination factor (LM factor and word insertion penalty).

L395 : highlighting the architecture’s strong potential when accurate localization is provided : This is a classic argument: if we had a good line segmenter, the recogniser would be good. However, the difficulty lies in achieving accurate segmentation, and difficult documents often present challenges for both segmentation and recognition.

---

### Author Response · Authors · 2025-12-02
**Meta Rebuttal: Consolidated Response to Reviewers**

We sincerely thank all reviewers for their thorough evaluations and constructive feedback. Below we address all major concerns across the four reviews.

---

## 1. **Novelty and Contribution** [to Reviewers PNbe, BcVz, aiU9]

**Core Innovation**: Our novelty lies in addressing a critical, previously unsolved gap. Pre-revolutionary Russian (19th-early 20th century) represents a genuinely low-resource domain with **_no prior HTR systems or comprehensive datasets_**. Unlike English (stable since 18th century), Russian underwent radical 1918 orthographic reform eliminating obsolete characters (ѣ, ѳ, ѵ, і), preventing direct application of modern Russian HTR models.

**Prior Work**: Existing solutions target narrow domains - Digital Peter (early 18th-century), Kokorin et al. (single author). Our system is **the first generalizable approach** across multiple 19th–20th century document types, enabling access to approximately 95% of undigitized Russian archival materials.

## 2. **Methodology & Baselines** [to Reviewers PNbe, BcVz, ekgW]

**Architecture**: We acknowledge using established components (YOLOv8 + TrOCR_pre). The contribution is **successful adaptation** for this unique domain, demonstrating 53.8% CER improvement over CNN-RNN baselines (18.4%→8.5%).

**Baseline Validity**: Direct comparison with SOTA models (LLMs, commercial OCR, DAN) is methodologically invalid - modern systems lack obsolete Cyrillic characters; DAN requires page-level annotation vs. our line-level workflow; cross-period evaluation would confound results due to ~150-year paleographic gap. We compare **architectures trained on our corpus** - the only valid approach.

## 3. **Dataset & Reproducibility** [to Reviewers PNbe, BcVz, aiU9]

Immediate access available to specialists for non-commercial research via [Anonymous request form](https://forms.yandex.com/cloud/6930757995add5edd8b99003). We are finalizing publication procedures with archival institutions: 38,501 annotated lines represents **the largest corpus** for pre-revolutionary Russian, spanning diverse scribes from 20+ provinces.

## 4. **Experimental Rigor** [to Reviewers PNbe, BcVz, ekgW]

**Out-of-Distribution**: Cross-archive testing is impossible - this is the first publicly annotated dataset for this period. We demonstrate multi-scribe generalization and are creating new manuscripts from other archives for future benchmarks.

**Component Analysis**: YOLOv8 achieves mAP=0.98, mean IoU 97.2%; segmentation-to-recognition gap: 1.3pp CER (7.2% with GT boxes vs. 8.5% full pipeline); TrOCR_pre handles slanted/curved lines through diverse geometry exposure during training.

## 5. **Technical Clarifications** [to Reviewers PNbe, aiU9, ekgW]

**Preprocessing**: Empirical evaluations confirm consistent gains. Our degraded historical manuscripts (ink deterioration, variable contrast) demonstrably benefit.

**Component-Wise Evaluation**: We acknowledge Tables 1-2 compare architectures and document types; a dedicated table illustrating phase-wise contributions within our YOLOv8+TrOCRpre pipeline would enhance clarity.

**Postprocessing**: Dictionary-based correction shows limited benefit; LLM-based approaches degrade performance (modern LLMs lack pre-reform orthography training data). Integrated language modeling with historical corpora is future work.

**YOLOv8 for Overlapping Lines**: We train YOLOv8 as joint detection-and-segmentation model. Mask-based cropping preserves original geometry; TrOCR robustly decodes non-horizontal text through grid-like feature maps.

## 6. **Performance & Usability** [to Reviewer ekgW]

**Accuracy**: WER 29.1% overall (50% on diaries) requires manual correction but represents **meaningful progress**. Many errors affect functional words/minor variants while preserving key entities and semantic structure.

**Document-Type Variance**: Performance correlates with formality - 4.8% CER (formal reports) vs. 19.0% CER (informal diaries) - reflecting task difficulty compounded by data scarcity (960 vs. 28,671 training samples).

## 7. **Paper Improvements** [to Reviewers PNbe, BcVz, aiU9]

- Add Appendix B: orthographic differences pre/post-reform Russian
- Clarify component-wise evaluation framework
- Refine "universal" scope (diverse styles within 19th–20th century, not all-period)
- Polish writing style to scientific paper standards
- Add complete baseline references and training details
- Correct YOLOv8 citation

---

### Meta-Review · Area_Chair_6HDN · 2026-01-06

**Summary:**

While the reviewers acknowledge the interest of tackling HTR for low-resource languages and of the proposed dataset, they express concerns about the technical originality of the proposed method, the comparisons to the state of the art, the limited evaluation (other datasets, ablation studies), and the presentation quality.

**Reviewer Concerns:**

The authors provided a general answer jointly covering the comments of all reviewers. Some of the specific concerns are convincingly covered, but the broader ones, such as limited technical originality and evaluation, remain open, with the authors' responses being unlikely to convince the reviewers.

**Reviewer Scores:**

Considering that the reviewers had reached a consensus towards rejection and that several broad concerns remain not entirely addressed, the AC expects that the reviewers would not have changed their scores.

---

### Decision · Program_Chairs · 2026-01-26

Reject